# Wearable Spectroradiometer for Dosimetry

**DOI:** 10.3390/s22228829

**Published:** 2022-11-15

**Authors:** Maximilian J. Chmielinski, Martin A. Cohen, Michael G. Yost, Christopher D. Simpson

**Affiliations:** Department of Environmental and Occupational Health Sciences, School of Public Health, University of Washington, Seattle, WA 98105, USA

**Keywords:** sensor applications, environmental monitoring, dosimetry

## Abstract

Available wearable dosimeters suffer from spectral mismatch during the measurement of broadband UV and visible radiation in environments that receive radiation from multiple sources emitting differing spectra. We observed this type of multi-spectra environment in all five Washington State cannabis farms visited during a field study investigating worker exposure to ultraviolet radiation in 2018. Spectroradiometers do not suffer from spectral mismatch in these environments, however, an extensive literature review conducted at the time of writing did not identify any spectroradiometers that were directly deployable as wearable dosimetry devices. To close this research gap, we developed a microcontroller system and platform that allows for researchers to mount and deploy the Ocean Insight Flame-S Spectroradiometer as a wearable device for measurement of UV and visible wavelengths (300 to 700 nm). The platform validation consisted of comparing measurements taken under platform control with measurements taken with the spectrometer controlled by a personal computer running the software provided by the spectroradiometer manufacturer. Three Mann–Whitney U-Tests (two-tailed, 95% CI), one for each intensity condition, compared the central tendency between the total spectral power (TSP), the integral of a spectrum measurement, measured under both control schemas. An additional analysis of per pixel agreement and overall platform stability was performed. The three Mann–Whitney tests returned no significant difference between the set of TSPs for each filter condition. These results suggest that the spectroradiometer takes measurements of equivalent accuracy under both control schemas, and can be deployed as a wearable device for the measurement of wavelength resolved UV and visible radiation.

## 1. Introduction

Both natural sunlight and artificial lamp-generated radiation are commonly used in a variety of horticultural settings, including cannabis plants [1]. These optical sources can contain both visible and ultraviolet (UV) wavelengths; however, overexposure to ultraviolet radiation causes potential health risks. The most common UV overexposure injury is erythema [2,3], and the most severe erythema is caused by wavelengths between 280 and 315 nm (UVB) [2,4]. All UV wavelengths are classified as a Group 1 carcinogen by the International Agency for Research on Cancer (IARC) [5,6]. Exposure of the eye to UV is associated with photokeratitis and keratoconjunctivitis. Injury severity is influenced by eye motion, spectral profile, and angle of incoming radiation. UVB exposure has been associated with photokeratitis [7], whereas both UVB and UVC overexposure has been causally linked to incidence of keratoconjunctivitis [8]. Blue light (wavelengths ~380–550 nm) can cause photochemical damage to the retina of the eye [9], and may impact circadian rhythm, alertness and cognitive performance [10].

We previously observed that cannabis farm workers completed tasks under lamps that emitted UV radiation at four out of five cannabis farms in Washington State [11]. While these observations confirm worker exposure to lamp-emitted radiation, the severity and prevalence of agricultural worker overexposure to lamp-emitted radiation is unstudied. A contributing reason to this research gap is because available wearable dosimeters cannot accurately measure exposure to broadband UV and visible radiation in an environment that contains light from multiple sources with different spectra [12], a condition observed at every cannabis farm we visited.

This inaccuracy results from the fact that radiation exposure is the sum of irradiance at each specific wavelength, which is then integrated over the wavelength range and period of interest. Wavelength-specific weightings, called action spectra or hazard spectra, which quantify the wavelength dependence of damage caused by UV radiation, are then applied to the wavelength-specific irradiance values to quantify the level of hazard that is present, and to determine compliance with occupational exposure limits. Broadband sensors perform well when the sensors are calibrated to a spectrum of interest (e.g., exposure to sunlight) and the environment of interest contains only this spectrum of interest. However, measurement of spectra that differ from that of the sensor calibration may lead to large measurement errors, named spectral mismatch errors [13].

Spectroradiometers do not suffer from spectral mismatch because they can measure wavelength-resolved irradiance, and hence can accurately measure radiation exposure in environments containing multiple sources with differing spectra. However, an extensive literature review did not identify any spectroradiometers that were directly deployable as wearable dosimetry devices. The ability of a wearable device to directly measure an individual’s exposure to radiation is important, because some prior studies have shown that measurements of ambient irradiance using a fixed location device do not always provide reliable estimates of personal exposure [14]. The commercially available Ocean Insight Flame-S Spectroradiometer can measure UV and visible light with a spectrum spanning a broad range of wavelengths, but its design restricts its use to laboratory bench settings.

This manuscript describes the development and validation of a microcontroller system that allows for researchers to mount and deploy the Ocean Insight Flame-S Spectroradiometer as a wearable device for the accurate measurement of wavelength-resolved exposure to radiation across a broad range of UV and visible wavelengths (from 300 to 700 nm).

## 2. Materials and Methods

### 2.1. Platform

This platform uses the Flame-S Spectrometer with an attached CC-3-UV-S cosine corrector optical inlet (Ocean Insight, Orlando, FL, USA). The QP450-1-XSR optical fiber (~USD326) connects the optical inlet and the spectroradiometer [15]. Although the Flame-S Spectrometer can nominally detect wavelengths down to 210 nm, the standard version of this device suffers from spectral scatter during measurements of broadband radiation, reducing its effective wavelength measurement range to approximately 300–700 nm, depending on the spectra of the measured source.

This platform, illustrated in Figure 1, uses the Arduino Due microcontroller board [16] and the RS232 communications standard [17] to control the spectroradiometer. Additional peripherals include a HiLetgo ILI9341 2.8” thin-film-transistor (TFT) liquid crystal display (LCD) [18], four non-latching momentary buttons, a 64 GB FAT32 SD card, a DS3231 I2C real-time clock (RTC) [19], and a CONXWAN B02P 26,800 mAh USB power bank [20]. The platform is currently housed in a 22 cm × 17 cm × 11 cm Zulkit hinged waterproof electrical box [21]. The overall cost of the system is ~$4400 USD for the spectrometer, optical fiber and optical inlet, plus an additional $130 USD for the peripheral components to convert the spectrometer to a wearable device. As tested, the platform weighs 1592 g. Appendix A provides a platform wiring diagram.

### 2.2. Test Source

The test source used to validate the performance of the spectroradiometer was a TriSOL TSS-300 Class AA + A solar simulator (OAI Inc., Milpitas, CA, USA), located at the University of Washington Clean Energy Institute [22]. This source generates a 300 mm × 300 mm square of radiation between 350 and 1200 nm. The working range of the collimated beam is 0.762 m, and the spatial uniformity is better than ±1%. The emitted irradiance matches that of the ASTM 173 1.5 air mass (AM) solar spectrum to an accuracy of ±15% and a temporal stability of <2% [23]. Prior to testing, the device underwent its 15-min warm-up cycle.

### 2.3. Optical Bench

An optical bench holds the optical inlet to the spectrometer at orthogonal at approximately the center of the 300 mm × 300 mm illuminated region of the solar simulator at a distance of 0.2 m below the test source. Irradiance intensity was controlled by the installation of zero, one, or two UV-fused silica 50 mm 2.0 optical density neutral density (ND) filters (Thorlabs, Newton, NJ, USA) in front of the spectroradiometer optical inlet. The experimental set up is shown in Figure 2.

### 2.4. Control Software

The manufacturer-provided control software, OceanView (ver 1.6.7, Ocean Insight, Orlando, FL, USA) allows for a computer running a Windows 10 or 11 operating system to control the spectroradiometer on the benchtop via USB 2.0 communication. In contrast, the Arduino Due board controls the spectroradiometer by sending commands and receiving data via the serial port. The Arduino operates using four states, consisting of a platform startup state, a spectroradiometer check state, a standby state, and a spectrum capture state. Should any state fail, the Arduino stops operation and sends an error code to the Serial port and, if initialized, the LCD screen [24].

The platform startup state occurs automatically upon the Arduino connecting to power. In this state the Arduino begins serial communication and causes the initialization of all peripherals. Next, the spectroradiometer check state involves the Arduino sending a command to change the integration time set on the spectroradiometer from the factory default of from 10 ms to 100 ms. When the Arduino receives a successful receipt of an electronic acknowledgement, an ‘ACK’, from the spectroradiometer, the Arduino transitions to the standby state [24]. In the standby state, the Arduino loads the control functions for the user interface to the LCD screens. Each control function is linked to one of the four momentary contact buttons on the platform. Upon the user pressing a button, the Arduino executes the associated function and moves the device into either the standby, spectroradiometer check, or the spectrum capture state.

During the spectrum capture, the Arduino sends a sequence of commands to the spectroradiometer that cause it to take a spectrum measurement, send spectrum data to the Arduino serial port, and have the Arduino save the data to the micro-SD card. Upon completion of this state, the Arduino moves into the standby state. The full source code for the firmware is accessible at the following repository: https://github.com/mxchml/Flame-Spectrocontroller, accessed 1 August 2022 [24].

Both control schemas convert raw pixel signal to irradiance in µW/(nm × cm^2^) through an identical five-step computation consisting of a noise correction, a linearity correction, a pixel wavelength-bound correction, an integration time application, and a calibration factor application. These processes are described in the Appendix A of this submission.

#### 2.4.1. Experiment #1: Platform Monitoring Sessions

The purpose of experiment #1 was to investigate the stability of the spectroradiometer measurement over a sustained duration while under platform control. For this purpose, the technician performed six monitoring sessions with the spectroradiometer under platform control to generate six sets of measurement data. During the six sessions, two sessions had zero neutral density (ND) filters installed, two had one ND filter installed, and two had two ND filters installed. For each session, the spectroradiometer was activated and the integration time was allowed to stabilize. After stabilization, the spectroradiometer was allowed to run for an arbitrary duration greater than one minute. All measurements taken prior to when the platform reached a stabilized integration time were not included in the analysis.

The adjustment of integration time between filter conditions, as well as the arbitrary sampling time beyond one minute resulted in the six datasets each comprising a variable number of spectral measurements (*n* = 4 to 23).

#### 2.4.2. Data Processing—Experiment #1

Each monitoring session comprised of a set of spectra (*n* = 4 to 23), and all spectra underwent integration by wavelength between the first pixel with a signal above threshold (varied by measurement, minimum was 353.313 nm) and 700 nm. This converted each spectrum to a value of total spectrum power (TSP) in units of mW/cm^2^. The coefficient of variation (COV) and the percentage difference between the largest and smallest TSP, called the maximum percent power difference (MPPD), were calculated for each of the six sessions.

#### 2.4.3. Experiment #2: Single Measurements Using Both Control Schemas

A priori, we expect the spectra and signal intensities to be identical between the platform controller and the commercial product. However, differences between the two control schemes in the measured magnitude and/or spectrum shape may occur due to differences in integration time and/or the differential implementation of signal processing. Therefore, the purpose of experiment #2 is to validate agreement between measurements taken by under both control systems.

To generate a range of intensities at the detector, zero, one, or two ND filters were installed between the solar simulator and the spectroradiometer. For each number of installed filters, the spectroradiometer took a set of three measurements under platform control and a set of three measurements under OceanView control, resulting in a total of six sets that collectively contain eighteen measurements. The technician switched the control schema by manually unplugging the spectroradiometer from the PC and connecting it to the Arduino controller platform.

#### 2.4.4. Data Processing—Experiment #2

TSPs were calculated for each spectrum using the process described previously. Additionally, the average spectrum for each of the six sets was calculated by averaging the irradiance measured at each pixel.

### 2.5. Analysis

#### 2.5.1. Mann–Whitney U-Tests

Three Mann–Whitney U-Tests (two-tailed, 95% CL) compared the central tendency of the TSPs calculated from the platform-controlled measurements to the TSPs calculated from the OceanView-controlled measurements for the zero-, one-, and two-filter conditions. A two-tailed Mann–Whitney U-Test at the 95% confidence level compared the central tendency of the set of COVs from the experiment #1 (*n* = 6) to the set of COVs from experiment #2 (*n* = 6).

#### 2.5.2. Linear Regressions

For each of the three optical filter settings, a linear regression compared the average spectrum generated by the platform control with the average spectrum generated by OceanView control. The goodness of fit metric is the R^2^ coefficient, while the slope of the regression line indicates quantitative agreement between the two spectrometer control schema.

For each pixel that measured above threshold for all filter conditions (*n* = 999), a linear regression based on three datapoints was performed. These three datapoints corresponded to the three-filter conditions, and each regression compared a pixel’s average measurement under platform control to the average measurement under OceanView control. The 999 calculated slopes were plotted against the pixel wavelength to create a scatterplot visualizing how the agreement between platform/OceanView controlled measurements changes across wavelengths.

## 3. Results

### 3.1. Experiment #1

Table 1 presents the stability data generated from the six platform-monitoring sessions and includes the number of installed filters, the number of samples, the duration of stable integration time measurement period, the COV, and the MPPD.

### 3.2. Experiment #2

Figure 3 presents six charts, displaying the average spectrum measured for each irradiance condition. (upper row: platform-controlled acquisition, bottom row: OceanView controlled acquisition). Each spectrum is the average of three individual measurements.

All the above plots begin at 353.313 nm. The lowest pixel irradiance measured above the threshold was 13.7, 0.19, and 0.018 µW/(nm × cm^2^) for the no-filter, one-filter and two-filter conditions under OceanView control and 15.6, 0.20, and 0.007 µW/(nm × cm^2^) for the no-filter, one-filter and two-filter conditions under platform control, respectively. Table 2 presents summary statistics for these measurements.

Table 2 contains the mean total power averages, standard deviations, coefficients of variation, and integration time statistics summarizing the results from experiment #2. Under OceanView control, the maximum TSP measured for a single spectrum was 40.3 during the no-filter condition and the minimum was 0.028 mW/cm^2^ under the two-filter condition. Under platform control, the maximum TSP measured for a single spectrum was 40.3 during the no-filter condition and the minimum was 0.028 mW/cm^2^ under the two-filter condition. The three Mann–Whitney tests returned no significant difference between the set of TSPs for each filter condition.

Figure 4 presents scatterplots and linear regression lines of the average spectra taken during OceanView control vs. average spectra taken during platform control.

The slopes and slope confidence intervals for the no-filter, one-filter, and two-filter regressions are 1.0117 ± 0.004, 0.9963 ± 0.005, and 1.1201 ± 0.008. The slope closest to unity came from the one-filter condition and had a value of 0.9963. The 2× filter regression resulted in the smallest y-intercept, with the intercept moving further from zero in the negative direction with each reduction in the number of filters present.

Figure 5 presents a plot of the slope of the linear regression calculated for each pixel’s average Oceanview vs. platform-controlled measurements.

The chart shows the regression slopes diverge from a value of unity at the wavelengths below approximately 370 nm, with the greatest divergence from unity occurring at 353 nm and having a value of 1.14.

## 4. Discussion

Each set of spectral measurements from experiment #1 had a low coefficient of variation and an MPPD that was below the solar simulator’s reported 2% temporal stability. Furthermore, each set of measurements from experiment #2 taken under platform control had a low coefficient of variation. These results demonstrate stable spectroradiometer measurement when under platform control.

The experiment #2 Mann–Whitney tests did not show a statistically significant difference in the TSPs between the platform- and OceanView-controlled measurement sets. These results suggest that the accuracy of the TSPs of the measurements taken when the spectroradiometer operates under platform control is comparable to those taken while the spectroradiometer operates under OceanView control.

The linear regression results of Figure 5 showed excellent agreement across all pixels, especially those corresponding to wavelengths above ~370 nm. Below ~370 nm, Figure 5 shows the slopes positively tailing as the scatterplot approaches 353 nm. This tail indicates that the spectroradiometer under platform control takes increasingly higher irradiance measurements when compared to those taken under OceanView control as the wavelength of measurement decreases. The suspected cause of this disagreement is that electrical and thermal noise contribute a greater proportion of the total signal at wavelengths below 370 nm, where the intensity of radiation emitted from the solar simulator source is low. Differences in the mean integration time may also have contributed to the differences in measurement noise between the control schema. However, the worst slope of 1.19 still indicates good agreement between the measurements under the two control schemas.

These results suggest that the accuracy of the spectra taken when the spectroradiometer operates under platform control are comparable to those taken while the spectroradiometer operates under OceanView control across all wavelengths between 353 nm and 700 nm. Agreement between measurements taken under both control schema suggest that the deployment of laboratory bench spectroradiometers using a simple lightweight control schema is a viable solution to the current dearth of available wearable spectroradiometric measurement technologies. Subsequently, we performed a comparison of instrument response to narrow band UVC radiation, under the two control schema. The radiation source for this test was an Ocean Optics mercury argon HG-2 calibration lamp. The difference in the measurement values was under 0.1% between the two control schema for all pixels between 253 and 255 nm. Additional testing with a broadband UVB and/or UVC optical source should be undertaken to further demonstrate the viability of this platform as a tool for measuring UVB and UVC exposure.

The usability of the device as a wearable sensor was evaluated by a Certified Industrial Hygienist (CIH) who manages a professional practice in the Washington DC area, as well as a second CIH working for the University of Washington field group [25]. As tested, the device is attached to a subject via a belt worn on the waist. The optical inlet was mounted on the frame of safety glasses, oriented to align with the subject’s eye. The optical fiber connecting the inlet to the housing ran down the subjects’ back, sandwiched between an undershirt and an overshirt to prevent the fiber from snagging on objects in the environment.

The DC area CIH reported that the device performed well under a variety of conditions and that the user interface was intuitive and allowed for him to operate the device without reference to a provided user manual. The UW CIH stated that the device was easy to use and comfortable to wear while walking; however, discomfort was reported when sitting while wearing the device. He suggested exploring whether chest-mounting strategies could improve comfort during sitting.

## Figures and Tables

**Figure 1 sensors-22-08829-f001:**
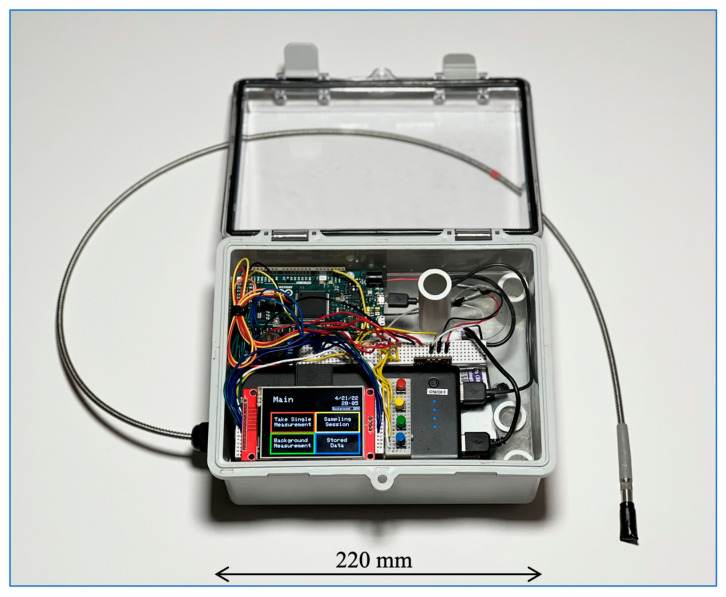
A top view of the wearable platform with hood open.

**Figure 2 sensors-22-08829-f002:**
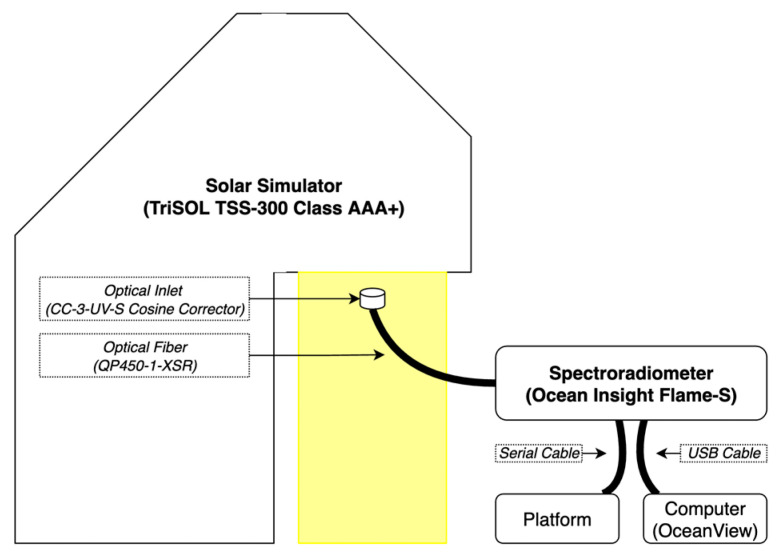
A simple block diagram of the testing setup.

**Figure 3 sensors-22-08829-f003:**
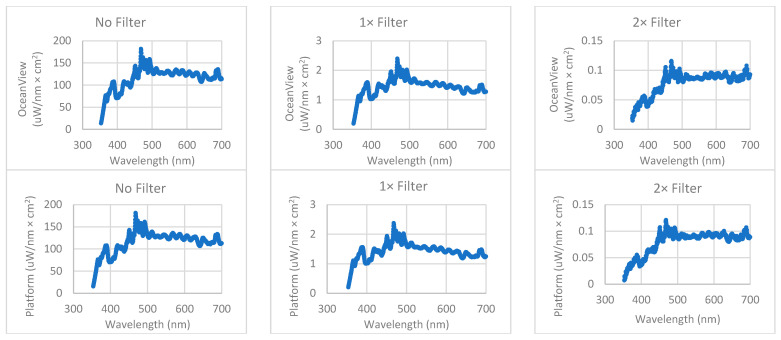
Each chart displays an average spectrum created by averaging three repeat spectral measurements. The top row shows the average spectrum of the platform-controlled measurements, and the bottom row shows the average spectrum of the OceanView controlled measurements.

**Figure 4 sensors-22-08829-f004:**
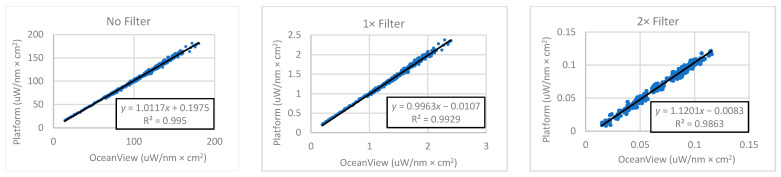
Scatterplots of measurements taken during OceanView control on the *x*-axis and platform control on the *y*-axis, both in units of µW/nm × cm^2^, for the three cycles. The plots also show the linear regression line, equation, and correlation coefficient.

**Figure 5 sensors-22-08829-f005:**
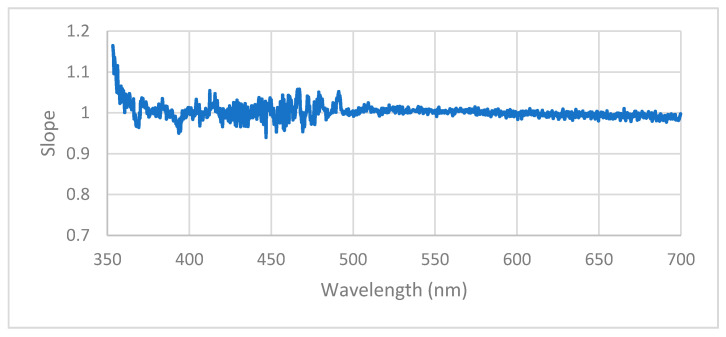
The slope of linear regressions for each pixel computed using three data points corresponding to platform- vs. OceanView-controlled measurement during each of the three filter conditions.

**Table 1 sensors-22-08829-t001:** Results from the platform monitoring sessions of the solar simulator emissions.

Filter	*n*	Stable Duration (s)	COV (%)	MPPD (%)
None	23	164	0.7	1.23
None	10	85	0.8	0.95
1×	11	98	0.2	0.20
1×	12	96	0.2	0.29
2×	4	100	0.1	0.11
2×	6	79	0.3	0.35

**Table 2 sensors-22-08829-t002:** The mean total power average, standard deviation, coefficient of variation, integration time (IT) mean and standard deviation, and the results of the Mann–Whitney test comparing the TSP for the OceanView vs. platform-controlled data.

Filter	OceanView						Mann–Whitney	Platform					
	*n*	Mean TSP (mW/cm^2^)	SD (mW/cm^2^)	COV	Mean IT (ms)	SD (ms)	U (*p* Value)	*n*	Mean TSP (mW/cm^2^)	SD (mW/cm^2^)	COV	Mean IT (ms)	IT SD (ms)
None	3	40.3	0.055	0.001	6.32	0	3 (0.663)	3	40.3	0.102	0.002	5.3	0.58
1×	3	0.503	0.001	0.002	484	0	0 (0.081)	3	0.497	0.001	0.001	452	2.65
2×	3	0.028	<0.001	0.014	3329	0	2 (0.383)	3	0.028	<0.001	0.010	5000	0

## Data Availability

All data referenced in this manuscript may be found at the following Github repository: https://github.com/mxchml/Flame-Spectrocontroller/blob/main/Solar%20Simulator%20Data%20and%20Analysis.xlsx.

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
