# Peer review of "Wearable Spectroradiometer for Dosimetry"

_sensors, 2022, doi:10.3390/s22228829_

Round 1

Reviewer 1 Report

In the present work, authors have developed a field deployable UV and visible radiation dosimeter (for 300 – 700 nm).   They have used a commercial spectroradiometer and added microcontrollers and other electronics, so as to use it as a wearable device.   Work is important, however, manuscript lacks novelty and reads more like a technical report than a research article.  Here are some points to consider to improve the manuscript.

1.  Literature survey can be more comprehensive.  For example, no literature references are provided for reported adverse health effects of UV radiation.   Similarly, what other UV sensors are being explored for dosimetry applications?

2.  Materials and methods:  A detailed block diagram of the setup shown in photograph (figure 1) can be provided.   Similarly, a figure to show the experimental setup along with light source will be useful.

3.  In experiment 1, how did authors arrive at n?  Why it is different for different filter configurations?        

4. In figure 2, spectra recorded using the commercial product and the platform developed are shown for different NDFs.  Both the results look same. Is there any time delay in the later?  More discussions are needed.  

5.  In page 8, usability of the developed platform in real time is discussed.  However, only mounting strategies for better comfort are discussed.   How the recorded spectrum looks in this case?  How it compares with the spectra recorded with the solar simulator?

6.  Finally, It is unclear what are the advantages (in terms of the recorded data) of the developed platform over commercial spectroradiomter is.  This point should be brought out clearly.

Reviewer 2 Report

In this manuscript, the authors developed a wearable spectroradiometer platform to measure the dosimetry for frontline farm workers. It is an interesting study and prototype. However, I have concerns in the novelty and scientific significance of this study. Here are some of my comments and suggestions:

1. The core of the whole system is the ocean optic spectrometer which is a commercial product. The control script was obtained from an online open source. To me, the authors are integrating everything together and characterizing a "mass production product" controlled by different softwares generating "similar" results. The novelty and significance of the study is not sufficient to me.

2. The authors mentioned the conventional dosimetry may be influenced by calibrations when testing different light sources. That's why all the dosimetry sensors have a designed detection wavelength range. In a broadband situation, people can use multiple sensors to generate the measurement results in a broader range. 

3. One of the purposes of using dosimetry sensors is they can be smaller, more sensitive, and they can be cheaper. Please add a scale bar in Figure 1. I think the size of the prototype is not so compact. It can be carried by a human for sure. But it won't be very light and comfortable.  Also, in the discussion part, it would be great if the authors can discuss the cost of the proposed method to conventional dosimetry sensors.

4. I am not so sure if the authors want to include that much statistical analysis in the main text. The physical spectrometer is the same. We probably don't need that much portion to show the readers the spectrometer performs the same under two different software.

5. In the introduction part, the authors mentioned one of the motivations is to help farm workers prevent exposing under strong UV radiation. That is an important application. However, no field tests or experiments have been conducted. The UVB and UVC part are not even fully studied. It would be great if the authors can include those part to show the capability.

Round 2

Reviewer 1 Report

Cost of various peripherals indicated in the revised manuscript is unnecessary.  Instead authors can make a statement on the final cost of commercial product and the system developed by them. 

Author Response

Reviewer Comment: Cost of various peripherals indicated in the revised manuscript is unnecessary.  Instead authors can make a statement on the final cost of commercial product and the system developed by them. 

Response: We have removed the itemized pricing and added a final cost statement.

Reviewer 2 Report

The authors response address most of my comments and concerns.

Author Response

Reviewer Comment: The authors response address most of my comments and concerns.

Response: Thank you, and we remain ready to provide any additional information upon request.